# First Report of *Stenotrophomonas maltophilia* from Canine Dermatological Infections: Unravelling Its Antimicrobial Resistance, Biofilm Formation, and Virulence Traits

**DOI:** 10.3390/antibiotics14070639

**Published:** 2025-06-23

**Authors:** Ria Rajeev, Porteen Kannan, Sureshkannan Sundaram, Sandhya Bhavani Mohan, Sivachandiran Radjendirane, Chaudhary Jeetendrakumar Harnathbhai, Anbazhagan Subbaiyan, Viswanathan Naveenkumar, Nithya Quintoil Mohanadasse, Wilfred Ruban Savariraj, Charley A. Cull, Raghavendra G. Amachawadi

**Affiliations:** 1Department of Veterinary Public Health and Epidemiology, Madras Veterinary College, Tamil Nadu Veterinary and Animal Sciences University (TANUVAS), Chennai 600051, India; riavet1997@gmail.com (R.R.); hodvphmvc@tanuvas.org.in (S.S.); scrvet2019@gmail.com (S.R.); 2Department of Clinics, Madras Veterinary College, Tamil Nadu Veterinary and Animal Sciences University (TANUVAS), Chennai 600051, India; sanjuri02@gmail.com; 3Department of Veterinary Public Health and Epidemiology, College of Veterinary Science and Animal Husbandry, Kamdhenu University, Anand 388001, India; drjitendra@kamdhenuuni.edu.in; 4ICMR-National Animal Resource Facility for Biomedical Research (NARFBR), Hyderabad 500101, India; anbazhagan.s@icmr.gov.in; 5Veterinary Clinical Complex, Veterinary College and Research Institute (VC&RI), Tamil Nadu Veterinary and Animal Sciences University (TANUVAS), Udumalpet 642205, India; naviviswanathan300@gmail.com; 6Department of Veterinary Public Health and Epidemiology, Rajiv Gandhi Institute of Veterinary Education and Research (RIVER), Puducherry 605009, India; nithyaquintoil.m@river.edu.in; 7Department of Livestock Products Technology, Karnataka Veterinary, Veterinary College, Animal and Fisheries Sciences University, Bengaluru 560024, India; rubanlpt@gmail.com; 8Midwest Veterinary Services, Inc., Oakland, NE 68045, USA; charley@mvsinc.net; 9Department of Clinical Sciences, College of Veterinary Medicine, Kansas State University, Manhattan, KS 66506, USA

**Keywords:** *stenotrophomonas maltophilia*, One Health, skin swabs, dogs, zoonotic

## Abstract

**Background/Objectives**: The present study was aimed at documenting *S. maltophilia* occurrence in dogs with skin ailments, investigating its virulence, biofilm-forming ability, antimicrobial susceptibility, and zoonotic potential to inform preventive and therapeutic strategies against multidrug resistant *S. maltophilia* infections. **Methods**: Skin swabs (*n* = 300) were collected from dogs with dermatological ailments. Isolation was performed using selective media and confirmed with molecular methods, validated by MALDI Biotyper. Antimicrobial susceptibility testing and efflux activity assessment were conducted. Resistance genes related to sulfonamides, quinolones, and β-lactams were screened. Virulence was assessed by biofilm formation, motility, and virulence gene profiling. **Results**: In total, 15 *S. maltophilia* (5%) isolates were identified. All 15 isolates were susceptible to trimethoprim-sulfamethoxazole, enrofloxacin, gatifloxacin, levofloxacin, minocycline, and tigecycline, but resistant to cefpodoxime and aztreonam. The following resistance genes *qnr* (93.3%), *bla*_OXA-48_ (46.7%), *bla*_KPC_ (33.3%), *bla*_NDM_ (33.3%), *bla*_CTX-M_ (20%), *bla*_SHV_ (20%), and *bla*_TEM_ (6.7%) were detected. All 15 isolates displayed high efflux activity. Overall, 9 isolates (60%) were strong biofilm producers, and 6 (40%) were moderate. Virulence genes such as *virB*, *motA*, *rmlA*, and *fliC* were present in all 15 isolates, with others varying in frequency. All isolates exhibited swimming motility. Heat map clustering showed diverse profiles, with no identical isolate patterns. Correlation analysis indicated positive associations between several antimicrobial resistance and virulence genes. **Conclusions**: This study underscores the zoonotic potential of *S. maltophilia* from dogs, advocating for a One Health approach to mitigate infection risks and limit the spread of virulent multidrug resistant pathogens.

## 1. Introduction

The genus *Stenotrophomonas*, belonging to the class Gammaproteobacteria, consists of non-fermenting gram-negative bacilli (NFGNB) that occur ubiquitously in environments such as soil, water, plant rhizospheres, and the bodies of animals and humans [1]. *Stenotrophomonas maltophilia* is the primary pathogen within this genus, implicated in various infections in humans. This opportunistic pathogen causes nosocomial infections, including respiratory, wound, urinary tract, ocular, and soft tissue infections, and ranks as the third most frequently isolated unusual NFGNB after *Pseudomonas aeruginosa* and *Acinetobacter baumannii* [2].

*S. maltophilia* expresses numerous virulence factors that facilitate host colonization, such as adhesins (lipopolysaccharides, flagella, type 1 fimbriae, type IV pili) and extracellular enzymes (proteases, esterases, lipases, gelatinase, hemolysin, siderophores, and cytotoxins) [3]. Its biofilm-forming ability on abiotic surfaces and host tissues diminishes the efficacy of drugs used to treat hospital-acquired infections [3]. Additionally, quorum sensing via autoinducers regulates gene expression based on cell density [4].

*S. maltophilia* is intrinsically resistant to a wide range of antibiotics, including β-lactams, fluoroquinolones, tetracyclines, chloramphenicol, and aminoglycosides. This resistance is mediated by two β-lactamases, L1 (a metallo-β-lactamase) and L2 (a clavulanic acid-susceptible cephalosporinase) [1]. Aminoglycoside resistance is also supported by modifying enzymes [AAC (6′)-Iz, APH (3′)-Iic, AAC (6′)-Iak] and efflux pumps (*SmeABC*, *SmeIJK*, *SmeYZ*) [1]. Additionally, resistance genes such as *sul1*, *sul2*, and *dfrA* associated with Class 1 integrons confer resistance to trimethoprim/sulfamethoxazole (TMP/SMX). TMP/SMX is the drug of choice for treating *S. maltophilia* infections, with alternate options like levofloxacin, minocycline, tigecycline, and ceftazidime used for TMP/SMX-resistant cases [2].

*S. maltophilia* has been isolated from animals with chronic respiratory disease and urinary infections, including horses, cats, dogs, and ball pythons [5,6]. However, only one study in India has reported its occurrence in animals, specifically in sarcoptic pig skin ulcers [7], with no prior studies in dogs. This study aimed to document *S. maltophilia* occurrence in dogs with skin conditions, evaluating its virulence, biofilm-forming capacity, and antimicrobial susceptibility to assess zoonotic potential and develop treatment strategies for biofilm-associated multidrug resistant *S. maltophilia* infections.

## 2. Results

### 2.1. Prevalence of S. maltophilia

A total of 15 isolates (5%, 15/300) showed characteristic colonies of *S. maltophilia* and were species-identified by morphological, biochemical, molecular (Figure 1), and MALDI-TOF MS methods (Table 1).

### 2.2. Characterization of Antimicrobial Resistance in S. maltophilia

All *S. maltophilia* isolates were susceptible to six antimicrobials: cotrimoxazole, enrofloxacin, gatifloxacin, levofloxacin, minocycline, and tigecycline. However, all isolates were resistant to cefpodoxime and aztreonam. Additionally, 14 isolates (93.33%) were resistant to imipenem and ceftriaxone, while 10 isolates (66.67%) exhibited resistance to piperacillin-tazobactum (Table 2). Molecular characterization revealed the presence of various ARGs in the isolates (Figure 2), with the *qnr* gene being most prevalent (93.33%). Other detected ARGs included *bla*_OXA-48_ (*n* = 7, 46.67%), *bla*_KPC_ (*n* = 5, 33.33%), *bla*_NDM_ (*n* = 5, 33.33%), *bla*_CTX-M_ (*n* = 3, 20%), *bla*_SHV_ (*n* = 3, 20%), and *bla*_TEM_ (*n* = 1, 6.67%) (Table 2). In addition to this, evaluation of the *S. maltophilia* isolates for bacterial pump efflux activity by the Etbr agar cartwheel method showed no fluorescence on the agar plates indicating active efflux activity (Figure 3) (Table 2).

### 2.3. Virulence Characterization of S. maltophilia

The crystal violet staining assay revealed that 9 of the *S. maltophilia* isolates (60%) were strong biofilm producers, while 6 isolates (40%) were moderate biofilm producers (Figure 4) (Table 2). Biofilm strength was categorized based on OD values as Negative: < 0.044; Weak: 0.044 < A ≤ 0.088; Moderate: 0.088 < A ≤ 0.176 and Strong: A > 0.176, where A-OD Value of Sample.

Virulence gene analysis revealed all the isolates harbored *virB* (100 %), *motA* (100 %), *rmlA* (100 %), and *fliC* (100 %) genes followed by *pilU* (93.3 %), *gspD* (80 %), *papD* (73.3 %), *hgbB* (73.3 %), *picNl* (66.6 %), *afaD* (53.3 %), and *hlyIII* (40 %) genes, respectively. However, the *frpC* and *fimH* genes were not found in any of the isolates (Table 2).

Swimming motility was assessed on modified LB plates, revealing that all 15 isolates exhibited swimming motility. Based on the diameter of the motility zone, 11 isolates (73.3%) were classified as strong, while 4 isolates (26.67%) were classified as moderate (Figure 5).

### 2.4. Association Between Antimicrobial Resistance, Virulence, Motility Pattern, and Biofilm-Forming Ability in S. maltophilia

The link between antimicrobial resistance, virulence, motility pattern, and biofilm-forming ability of *S. maltophilia* isolated from skin swabs of dogs with dermatological ailments was established by hierarchical clustering (dendrogram). The heatmap showed the grouping of isolates into mainly three clusters (A, B, and C) (Figure 6). Cluster A consisted of 4 isolates namely SM 13, SM 7, SM 1, and SM 6 which harbored *virB*, *motA*, *rmlA*, *fliC*, *pilU*, *hgbB*, *picNl*, and *hlyIII* virulent genes; *bla*_NDM_ ARG and resistant to cefpodoxime, aztreonam, imipenem, and ceftriaxone in common. Cluster B consisted of 5 isolates namely SM 10, SM 14, SM 9, SM 5, and SM 8 which harbored *virB*, *motA*, *rmlA*, *fliC*, *pilU*, *picNl*, and *gspD* virulent genes; *qnr* ARG and resistant to cefpodoxime, aztreonam, imipenem, piperacillin-tazobactum, and ceftriaxone in common. Cluster C consisted of 5 isolates namely SM 3, SM 12, SM 4, SM 11, and SM 15 which harbored *virB*, *motA*, *rmlA*, *fliC*, *pilU*, and *papD* virulent genes; *qnr* and *bla*_O_ ARGs and resistant to cefpodoxime, aztreonam, imipenem, and ceftriaxone in common. Isolate SM 2 remained outgroup. Interestingly, among the 15 isolates, none of the isolates harbored the same pattern of antimicrobial resistance, virulence, motility pattern, and biofilm-forming ability. Further association between antimicrobial resistance and virulence gene was established. A moderate positive correlation was observed between *bla*_SHV_ with *zot* and *tpsB* (+0.53 to +0.58), *bla*_TEM_ with *lktD* and *hcp* (+0.53 to +0.68), *bla*_OXA-48_ with *papD* (+0.56) and *qnr* with *gspD* (+0.53) (Figure 7). On the other hand, a weak negative correlation was seen between *bla*_KPC_ with *tpsB* (−0.35), *bla*_OXA-48_ with *plcN1* (−0.47), and *qnr* with *hylIII* and *lktD* (−0.33 to −0.53) (Figure 7).

## 3. Discussion

*S. maltophilia* has emerged as an important opportunistic pathogen which has been increasingly reported worldwide [1]. Although traditionally considered as a human pathogen, it has been reported in dogs, horses, cats, dogs, and ball pythons [5,6]. Earlier studies have reported that animal strains of *S. maltophilia* shared phylogenetic traits with some of the most successful human strains [8]. In this study, 15 *S. maltophilia* isolates were obtained from the skin swabs of dogs with dermatological ailments. These results indicate its involvement either as a primary pathogen or as part of a polymicrobial infection. Further studies would be necessary to confirm whether *S. maltophilia* is a direct cause of the dermatological problems or a secondary invader exploiting the diseased environment. This is the first study globally to document *S. maltophilia* from dogs with dermatological ailments.

An antibiotic susceptibility test revealed that the isolates were susceptible to cotrimoxazole, enrofloxacin, gatifloxacin, levofloxacin, minocycline, tigecycline, and ticarcillin-clavulanate. As fluoroquinolones and tetracyclines are commonly used in both human and veterinary medicine, the findings of this study support their potential use for treatment in cases involving *S. maltophilia* [2]. Furthermore, the isolates were found to be resistant to cefpodoxime, aztreonam, imipenem, ceftriaxone, and piperacillin-tazobactam. Carbapenem resistance is particularly worrisome, as they are considered a last-resort antibiotic for treating MDR infections [9]. A limitation of this study is the lack of specific CLSI breakpoints for *S. maltophilia* for several antibiotics, including enrofloxacin, gatifloxacin, imipenem, piperacillin-tazobactam, ceftriaxone, cefpodoxime, aztreonam, and tigecycline. In the absence of established breakpoints for *S. maltophilia*, we interpreted antimicrobial susceptibility results based on the available literature and expert recommendations, which may not fully reflect clinical susceptibility patterns. Further studies are needed to refine and standardize breakpoints for *S. maltophilia* to improve the accuracy and relevance of antimicrobial resistance data for this pathogen.

The resistance to β-lactams and carbapenems might be attributed to intrinsic resistance mechanisms typical of *S. maltophilia* or acquired resistance genes [10]. The molecular characterization of ARGs in *S. maltophilia* isolates demonstrates a high prevalence of the *qnr* gene (93.33 %) and a concerning presence of multiple β-lactamase genes, including *bla*_OXA-48_ (46.67 %), *bla*_KPC_ (33.33%), *bla*_NDM_ (33.33%), *bla*_CTX-M_ (20%), *bla*_SHV_ (20%), and *bla*_TEM_ (6.67%). There was a broad spectrum of β-lactamase enzymes, capable of hydrolyzing a wide range of β-lactam antibiotics, including carbapenems, cephalosporins, and penicillins, which are often considered last-resort treatments for MDR infections. While this represents the first investigation of *S. maltophilia* specifically from canine dermatological sources, our findings can be meaningfully contextualized within the broader literature. Our detection of *qnr* (93.3%) aligns with previous studies reporting a high prevalence of quinolone resistance mechanisms in *S. maltophilia* clinical isolates [11,12]. The presence of multiple β-lactamase genes including *bla*_OXA-48_ (46.7%), *bla*_KPC_ (33.3%), and *bla*_NDM_ (33.3%) is consistent with emerging trends of carbapenemase-producing *S. maltophilia* in healthcare settings [13,14,15], though the co-occurrence of multiple resistance genes in our canine isolates suggests potential horizontal gene transfer mechanisms warranting further investigation.

Over-expression of the bacterial efflux pump systems is one of the mechanisms by which bacteria exhibit multi-drug resistance [16]. All the isolates showed high efflux activity resulting in MDR phenotype. These findings are in line with the AST results and presence of ARGs which showed that the 15 isolates were resistant to more than 3 antibiotics tested and 14 out of 15 isolates harbored multiple ARGs. These findings point to a complex resistance profile that complicates treatment options for infections caused by these isolates. Therefore, targeted therapeutic strategies are crucial for understanding the genetic basis of resistance and managing the spread of resistant bacteria in both animal and human populations.

Among *S. maltophilia* virulence factors, biofilm plays an important role in the survivability and virulence. In this study, all *S. maltophilia* isolates were able to produce biofilm and most of them were strong biofilm producers (60%) followed by moderate biofilm producers (40%) which is comparable to rates reported in clinical *S. maltophilia* isolates from human infections, where biofilm formation rates typically range from 70 to 90% [17,18,19]. The strong biofilm-forming isolates identified in this study are likely to exhibit increased resistance to standard treatments, as biofilms create a physical barrier that limits the penetration of antibiotics and reduces their efficacy. The slightly lower rates in our study may reflect source-specific variations or environmental factors affecting canine skin isolates. In the study of motility, most of the isolates exhibited strong or moderate levels of swimming patterns; however, none of the isolates were able to swarm which was in line with previous findings [20,21].

The screening of virulence genes in *S. maltophilia* showed a high prevalence of *virB*, *motA*, *rmlA*, and *fliC* in 100% of the isolates. Gene *virB* encodes components of the Type IV secretion system (T4SS), which is involved in the translocation of effector proteins into host cells, contributing to host cell manipulation and immune evasion [22]. Furthermore, *motA* and *fliC* are associated with motility, colonization, and biofilm formation, making them essential for bacterial persistence in diverse environments [3]. Also, *rmlA* is responsible for biosynthesis of rhamnose, a sugar required for the production of lipopolysaccharides (LPS) in the bacterial outer membrane, which plays a role in immune evasion and biofilm formation [23]. Moderate prevalence of *pilU* (93.3%) encoding for type IV pili, which is essential for bacterial adherence to host tissues and surfaces and translocation; *gspD* (80%) encoding a component of the Type II secretion system (T2SS); *papD* (73.3%) involved in the assembly of P pili associated with bacterial adherence; *hgbB* (73.3%) encoding for a protein involved in iron acquisition, which is crucial in iron-limited environments; and *plcN1* (66.6%) responsible for bacterial immune evasion and tissue invasion, further emphasizing the bacterium’s ability to establish infections in various host environments were recorded [3,22,23]. The lower prevalence of virulence genes such as *afaD* (53.3%) and *hlyIII* (40%) encoding for adhesins and haemolysin protein were observed among the isolates.

The presence of multiple virulence genes, particularly those related to secretion systems (T4SS and T2SS), motility, adhesion, and iron acquisition, paints a picture of *S. maltophilia* as a highly adaptable pathogen capable of surviving in diverse environments and host tissues. These virulence factors likely contribute to the bacterium’s ability to persist in chronic infections, form biofilms, and resist host immune responses. The high prevalence of *motA* and *fliC* suggests that motility is a critical trait for environmental colonization and infection initiation, while the presence of secretion systems (*virB* and *gspD*) underscores the bacterium’s capacity to secrete effector molecules that manipulate host cells and degrade tissues [22].

A heat map was established to study the link between antimicrobial resistance, virulence, motility pattern, and biofilm-forming ability of *S. maltophilia* which showed that the isolates have a clonal connection. The hierarchical clustering exhibited diverse profiles with no two isolates sharing the same pattern of traits. This could be attributable to the adaptive versatility of *S. maltophilia*, which enables it to thrive in different environmental niches and cause various infections. These clusters suggests that the pathogenic potential of *S. maltophilia* may vary significantly between isolates, even when they are from same source.

The correlation matrix analysis revealed interesting associations between ARGs and virulence factors, suggesting potential co-regulation or functional linkages between these traits. The moderate positive correlation between *bla*_SHV_, a β-lactamase gene conferring resistance to β-lactam antibiotics, and the virulence genes *tpsB* (Type VI secretion system) and *zot* (zonula occludens toxin) (+0.53 to +0.58), suggests that some resistance mechanisms may be linked with the bacterium’s ability to deliver virulence factors to host cells. The association between *bla*_TEM_ and *lktD* (leukotoxin D) and *hcp* (Type VI secretion system protein) (+0.53 to +0.68) points to a possible synergy between resistance to β-lactams and the secretion of toxins or immune-modulating proteins. Similarly, *bla*_OXA-48_ a carbapenemase gene, correlates positively with *papD* (+0.56), a gene involved in the assembly of pili, which is important for bacterial adhesion to host surfaces suggesting that increased resistance to carbapenems may enhance the bacterium’s capacity to adhere to host tissues, potentially making infections harder to treat. The positive correlation between the quinolone resistance gene *qnr* and *gspD* (Type II secretion system) suggests that resistance to fluoroquinolones may be linked with the bacterium’s ability to secrete enzymes and toxins, further complicating treatment. A weak negative correlation was seen between *bla*_KPC_ with *tpsB* (−0.35), *bla*_OXA-48_ with *plcN1* (−0.47) and *qnr* with *lktD* and *hylIII* (−0.33 to −0.53) suggesting that resistance of said genes are less dependent on certain virulence mechanisms. The correlation between antimicrobial resistance and virulence genes observed in our study echoes findings from human clinical isolates, where similar associations between resistance mechanisms and pathogenicity factors have been documented [24,25]. This suggests that the pathogenic potential and resistance profiles of *S. maltophilia* may be conserved across different host species, supporting the zoonotic concern we highlight.

## 4. Materials and Methods

### 4.1. Isolation and Characterization of S. maltophilia

#### 4.1.1. Sample Collection

The present study was carried out from January 2024 to March 2024 in the Teaching Veterinary Clinical Complex, Madras Veterinary College, Chennai. A total of 300 skin swabs were collected aseptically from dogs presented with skin ailments and transported to the Bacteriology Laboratory of the Department of Veterinary Public Health and Epidemiology for further processing.

#### 4.1.2. Bacterial Isolation, Identification, and Molecular Confirmation

Skin swabs were inoculated in 5 mL Luria Bertani Broth and incubated at 37 °C for 24 h. The enriched inoculum was then streaked onto Stenotrophomonas selective agar with vancomycin (5 mg), imipenem (32 mg), and amphotericin B (2.5 mg) and incubated overnight at 37 °C. Characteristic *S. maltophilia* colonies appeared as dark green with a blue halo. Presumptive colonies underwent gram staining, oxidase, catalase, and glucose fermentation tests for confirmation. PCR targeting species-specific 23S rRNA [26] was used for molecular characterization, following a 25 µL reaction protocol with a specific thermal cycle (Table 3). Optimization of PCR was carried out using reference strain *S. maltophilia* (MCC2083) obtained from the National Centre for Microbial Resource, Pune, India.

#### 4.1.3. Matrix-Assisted Laser Desorption Ionization–Time of Flight Mass Spectrometry (MALDI-TOF MS) Identification

Bacterial isolates were prepared for MALDI-TOF MS following the ethanol-formic acid extraction method according to the manufacturer’s instructions. A loopful of bacteria was suspended in 300 µL distilled water with 900 µL ethanol, centrifuged at 17,000× *g* for 2 min, and the supernatant discarded. After repeated centrifugation and ethanol removal, the pellet was air-dried and resuspended in 5–50 µL of formic acid–water (70:30). An equal volume of acetonitrile was added, centrifuged, and 1 µL of the supernatant was placed on the MALDI target plate. After drying, it was overlaid with 1 µL matrix solution (α-Cyano-4-hydroxycinnamic acid). Mass spectra were analyzed on a Microflex LT mass spectrometer (Bruker) with Biotyper software (V 3.0) using Bruker bacterial test standard for identification.
antibiotics-14-00639-t003_Table 3Table 3List of primers used for the detection of *S. maltophilia*.GenePrimer SequenceCyclic ConditionNo. of CyclesAmplicon Size (bp)ReferenceStepTemperature and Time**Species specific PCR****23S rRNA**F-GCTGGATTGGTTCTAGGAAAACGCR-ACGCAGTCACTCCTTGCGInitial denaturation94 °C–5 min
278[26]Denaturation94 °C–45 s30Annealing68 °C–45 sExtension72 °C–45 sFinal extension72 °C–10 min
**Antimicrobial resistance genes*****bla_SIM_***F-GTACAAGGGATTCGGCATCGR-TGGCCTGTTCCCATGTGAGInitial denaturation 95 °C–4 min
569[27]Denaturation95 °C–45 s30Annealing58 °C–60 s***bla_VIM_***F-GTTTGGTCGCATATCGCAACR-AATGCGCAGCACCAGGATAG382Extension72 °C–40 sFinal Extension72 °C–5 min
***bla*_TEM_**F-TCCGCTCATGAGACAATAACCR-TTGGTCTGACAGTTACCAATGCInitial denaturation 94 °C–5 min
931[28]Denaturation 94 °C–30 s35Annealing60 °C–15 s***bla*_CTX-M_**F-ATGTGCAGYACCAGTAARGTKATGGCR-TGGGTRAARTARGTSACCAGAAYCAGCGGExtension72 °C–30 s593Final Extension72 °C–5 min
***bla*_SHV_**F-AGCCGCTTGAGCAAATTAAACR-ATCCCGCAGATAAATCACCACInitial denaturation 95 °C–5 min
713[29]Denaturation 94 °C–45 s35Annealing53 °C–45 sExtension72 °C–1 minFinal Extension72 °C–10 min
***bla*_KPC_**F-CGTCTAGTTCTGCTGTCTTGR-CTTGTCATCCTTGTTAGGCGInitial denaturation 94 °C–10 min
798[30]Denaturation 94 °C–30 s30Annealing52 °C–40 sExtension72 °C–50 sFinal Extension72 °C–5 min
***bla*_NDM_**F-TCGCATAAAACGCCTCTGR-GAAACTGTCGCACCTCATInitial denaturation 95 °C–6 min
1001[31]Denaturation 95 °C–45 s32Annealing55 °C–45 sExtension72 °C–60 sFinal Extension72 °C–7 min
***qnr***F-ACACAGAACGGCTGGACTGCR-TTCAACGACGTGGAGCTGTInitial denaturation 95 °C–5 min
817[32]Denaturation 95 °C–60 s30Annealing55 °C–60 sExtension68 °C–60 sFinal Extension68 °C–5 min
***Sul1***F-TAGCGAGGGCTTTACTAAGCR-ATTCAGAATGCCGAACACCGInitial denaturation95 °C–5 min
437[33]Denaturation95 °C–1 min35Annealing55 °C–60 s***Sul2***F-CCTGTTTCGTCCGACACAGAR-GAAGCGCAGCCGCAATTCAT956Extension72 °C–1 minFinal Extension72 °C–10 min
***bla*_OXA-48_**F-GCGTGGTTAAGGATGAACACR-CATCAAGTTCAACCCAACCGInitial denaturation 95 °C–5 min
438[34]Denaturation 95 °C–45 s35Annealing60 °C–45 sExtension72 °C–1 minFinal Extension72 °C–8 min
**Virulence genes*****entA***F: CGTTCGCACTCGACGTGACR: CGAACTGACGGTAACGATCACGInitial denaturation 94 °C–5 min
251[35]Denaturation 94 °C–30 s34Annealing60 °C–30 sExtension72 °C–30 sFinal Extension72 °C–5 min
***stmPr1***F: TGAAAGCAAATGCGCCGTTGR: GTGATGGCGTCGGTGATGTCInitial denaturation 94 °C–5 min
852Denaturation 94 °C–30 s34Annealing60 °C–30 sExtension72 °C–30 sFinal Extension72 °C–5 min
***hlyIII***
F: CGTCCATTGCTTCGATCCGTGR:
GACGAAGTGGCAGACGCTG
Initial denaturation 94 °C–5 min
607Denaturation 94 °C–30 s34Annealing60 °C–30 sExtension72 °C–30 sFinal Extension72 °C–5 min
***fimH***
F: GATCCGCCTGAACTGCCAGR:
CTGGCAGTTCAGGCGGATC
Initial denaturation 94 °C–5 min
576Denaturation 94 °C–30 s34Annealing60 °C–30 sExtension72 °C–30 sFinal Extension72 °C–5 min
***hgbB***F: GGACATCCAGAACATGGGTGCR: GGATCGATCGTGTACGGACCInitial denaturation 94 °C–5 min
1239Denaturation 94 °C–30 s34Annealing60 °C–30 sExtension72 °C–30 sFinal Extension72 °C–5 min
***virB***F: GCATCATGCAGAACGAGCTGR: GACGGCTCGTACTTCTGCACInitial denaturation 95 °C–5 min
1075Denaturation 95 °C–45 s35Annealing60 °C–45 sExtension72 °C–1 minFinal Extension72 °C–8 min
***frpC***F: CCAGTTCAACCTGTCGATGCTGR: CACCGAACAGGTTGTCCCAGInitial denaturation 95 °C–5 min
653Denaturation 95 °C–45 s35Annealing60 °C–45 sExtension72 °C–1 minFinal Extension72 °C–8 min
***afaD***F: GAAGCGCCTGACTGCCTTTTGR: GATCACGTTGTAAGGCCGCCInitial denaturation 95 °C–5 min
328Denaturation 95 °C–45 s35Annealing60 °C–45 sExtension72 °C–1 minFinal Extension72 °C–8 min
***fhaB***F: GTATCGCACAACCGCTTCCAGR: CGTCGTTGATGACCTTCTGCACInitial denaturation 95 °C–5 min
1744Denaturation 95 °C–45 s35Annealing60 °C–45 sExtension72 °C–1 minFinal Extension72 °C–8 min
***papD***F: CACGCGAGTGATCTATCCGGR: GTGATGAAGCGCACCTGGTCInitial denaturation 95 °C–5 min
579Denaturation 95 °C–45 s35Annealing60 °C–45 sExtension72 °C–1 minFinal Extension72 °C–8 min
***gspD***F: GTCGACACCGATATCGGTGGR: GGTAGACCACATGCAGGTTGCInitial denaturation 94 °C–5 min
694Denaturation 94 °C–30 s32Annealing60 °C–15 sExtension72 °C–30 sFinal Extension72 °C–5 min
***hcp***F: GACGGCAACGCGATCAATTACR: GTTCTTGGTTGCACTCCACTGInitial denaturation 94 °C–5 min
201Denaturation 94 °C–30 s32Annealing60 °C–15 sExtension72 °C–30 sFinal Extension72 °C–5 min
***zot***F: GCGTCAGTACACCGATGGTTGR: GCAGGCAGTGTCCAGCATGInitial denaturation 94 °C–5 min
431Denaturation 94 °C–30 s32Annealing60 °C–15 sExtension72 °C–30 sFinal Extension72 °C–5 min
***plcN1***F: GTGACCGATATCGGCCGACR: CTGGAAGTGGCGGTGGAAGInitial denaturation 94 °C–5 min
1779Denaturation 94 °C–30 s34Annealing62 °C–30 sExtension72 °C–30 sFinal Extension72 °C–5 min
***pilU***F: CGACCACCATCGATTTCACTTCGR: GACAGGTCCATCAGCAGCTGInitial denaturation 94 °C–5 min
778Denaturation 94 °C–30 s34Annealing60 °C–30 sExtension72 °C–30 sFinal Extension72 °C–5 min
***fliC***F: CGATCTCCGAGCGCTTCGR: GAACAGCTGGCTGGAGAACGInitial denaturation 94 °C–5 min
296Denaturation 94 °C–30 s34Annealing60 °C–30 sExtension72 °C–30 sFinal Extension72 °C–5 min
***rmlA***F: CTCAGCGTGCTGATGCTGGR: GATGAAGTTGGAGGCTTCCAGCInitial denaturation 94 °C–5 min
600Denaturation 94 °C–30 s34Annealing60 °C–30 sExtension72 °C–30 sFinal Extension72 °C–5 min
***tpsB***F: GTGGACATCGTGATGAAGCGCR: CTTGCCGATGAAGTGACGGTGInitial denaturation 94 °C–5 min
822Denaturation 94 °C–30 s34Annealing54 °C–30 sExtension72 °C–30 sFinal Extension72 °C–5 min
***motA***F: CGTTGGATTCCTGGTCGTCATCR: GAGCCCATGGTGATGACGATGInitial denaturation 94 °C–5 min
558Denaturation 94 °C–30 s34Annealing54 °C–30 sExtension72 °C–30 sFinal Extension72 °C–5 min
***lktD***F: GCACATCCGTGATGCAGTCGR: CGAGATTCTCGTCCTGCATGGInitial denaturation94 °C–5 min
1235Denaturation 94 °C–30 s34Annealing54 °C–30 sExtension72° C–30 sFinal Extension72 °C–5 min


### 4.2. Screening of the Isolates for the Presence of Antimicrobial Resistance

#### 4.2.1. Antimicrobial Susceptibility Testing (AST)

All isolates underwent antimicrobial sensitivity testing using the Kirby–Bauer disc diffusion method against 12 antibiotics: trimethoprim-sulfamethoxazole (23.75 µg/1.25 µg), enrofloxacin (5 µg), gatifloxacin (5 µg), levofloxacin (5 µg), imipenem (30 µg), piperacillin-tazobactam (100 µg/10 µg), ticarcillin-clavulanate (75 µg/10 µg), ceftriaxone (30 µg), cefpodoxime (10 µg), aztreonam (30 µg), minocycline (30 µg), and tigecycline (15 µg). Isolates were grown in Luria Bertani Broth at 37 °C for 16 h, spread on Mueller–Hinton agar plates, and allowed to dry. Antibiotic discs were placed on the plates within 15 min, and plates were incubated overnight at 37 °C. Zones of inhibition were measured and classified as sensitive, intermediate, or resistant per CLSI guidelines [36].

#### 4.2.2. Assessment of Bacterial Efflux Pump Activity in *S. maltophilia*

Efflux pump activity in *S. maltophilia* was assessed using the ethidium bromide agar cartwheel method [16]. Trypticase Soy agar plates with ethidium bromide (0.0–2.5 mg/L) were divided into 12 radial sectors. Bacterial cultures (0.5 McFarland) were swabbed from center to edge and incubated at 37 °C for 16 h. Fluorescence was then observed under a gel documentation system.

#### 4.2.3. Antimicrobial Resistance Genes (ARGs)

Screening of antimicrobial genes in *S. maltophilia* was carried out by PCR with cyclic conditions as described in Table 3. The ARGs targeted were *bla*_CTX-M_ and *bla*_TEM_ [28], *bla*_SHV_ [29], *bla*_OXA-48_ [34], *bla*_KPC_ [30], *bla*_NDM_ [31], *bla*_VIM_, and *bla*_SIM_ [27] encoding for β-lactam resistance, *qnr* [32] encoding for quinolone resistance, and *sul-1* and *sul-2* [33] encoding for sulphonamide resistance.

### 4.3. Characterization of the Virulence Properties of S. maltophilia

#### 4.3.1. Evaluation of Biofilm-Forming Ability

Biofilm-forming ability of isolates was assessed by crystal violet staining [37,38]. Cultures (1.0 McFarland, diluted 1:100) were incubated in microtiter plates at 37 °C for 24 h. After washing and fixing, biofilms were stained, destained, and quantified by OD at 492 nm. Biofilm strength was categorized based on OD values as Negative: < Ac; Weak: Ac < A ≤ 2Ac; Moderate: 2Ac < A ≤ 4Ac, and Strong: A > 4Ac, where Ac-OD Value of Negative control; A-OD Value of Sample.

#### 4.3.2. Motility Assay

Swimming motility of *S. maltophilia* was assessed on modified LB agar (0.3% w/v) plates [20]. In total, 10 µL of cultures (0.5 McFarland) were inoculated at the center and incubated at 37 °C for 24 h. Halo diameters were recorded as high (>10 mm), moderate (>7 to ≤10 mm), or low motility (> 6 to ≤7 mm) [35].

#### 4.3.3. Molecular Characterization of Virulence Genes

The screening of 20 virulence genes in *S. maltophilia* isolates was conducted through multiplex PCR which included set I (*entA*, *stmPr1*, *hlyIII*, *fimH*, *hgbB*), set II (*virB*, *frpC*, *afaD*, *fhaB*, *papD*), set III (*gspD*, *hcp*, *zot*), set IV (*pilU*, *fliC*, *rmlA*), set V (*tpsB*, *motA*, *lktD*), and *picN1* as described by [35] (Table 3).

### 4.4. Statistical Analysis

A heatmap with hierarchical clustering was generated to visualize the distribution of antimicrobial resistance, virulence, motility, and biofilm formation in *S. maltophilia* isolates [39]. Spearman’s correlation between resistance and virulence genes was analyzed in R (Version 4.3.1) using pheatmap and rcorr packages, with significance at *p* < 0.05. Data visualization employed Origin 2024 (trial version).

## 5. Conclusions

In summary, our study documented the presence of multidrug-resistant virulent *S. maltophilia* isolates in dogs with dermatological ailments at the Teaching Veterinary Hospital, Madras Veterinary College, India. It highlights significant genetic diversity among isolates and is the first report of *S. maltophilia* in dogs from India. The varying profiles of antimicrobial resistance, virulence gene expression, motility, and biofilm-forming ability emphasize the need for vigilant antimicrobial stewardship in veterinary settings. The potential for zoonotic transmission from infected pets necessitates a One Health approach to mitigate risks associated with this opportunistic pathogen and promote health in both humans and animals.

## Figures and Tables

**Figure 1 antibiotics-14-00639-f001:**
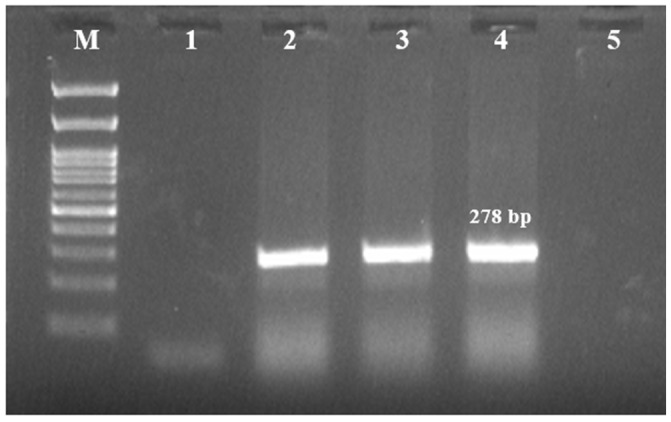
PCR amplification of 23S rRNA gene of *S. maltophilia*. M—100 bp ladder; 1—No template control; 2—Positive control (*S. maltophilia*–MCC2083); 3, 4—Positive samples; 5—Negative sample.

**Figure 2 antibiotics-14-00639-f002:**
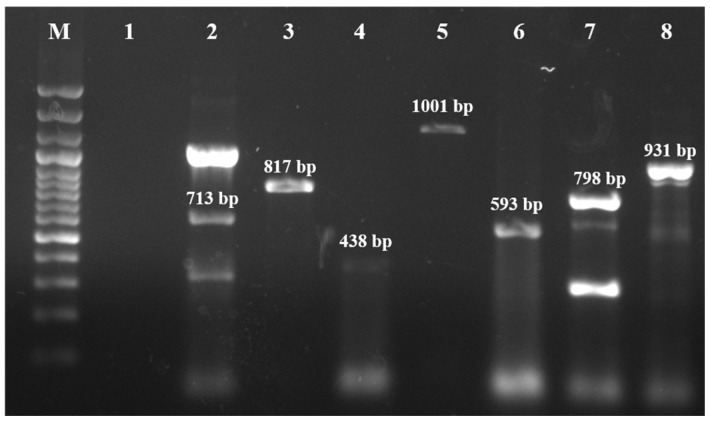
PCR amplification of AMR genes in *S. maltophilia*. M—100 bp ladder; 1—No template control; 2—*bla*_SHV_ gene (713 bp); 3—*qnr* gene (817 bp); 4—*bla*_OXA-48_ gene (438 bp); 5—*bla*_NDM_ gene (1001 bp); 6—*bla*_CTX-M_ gene (593 bp); 7—*bla*_KPC_ gene (798 bp); 8—*bla*_TEM_ gene (931 bp).

**Figure 3 antibiotics-14-00639-f003:**
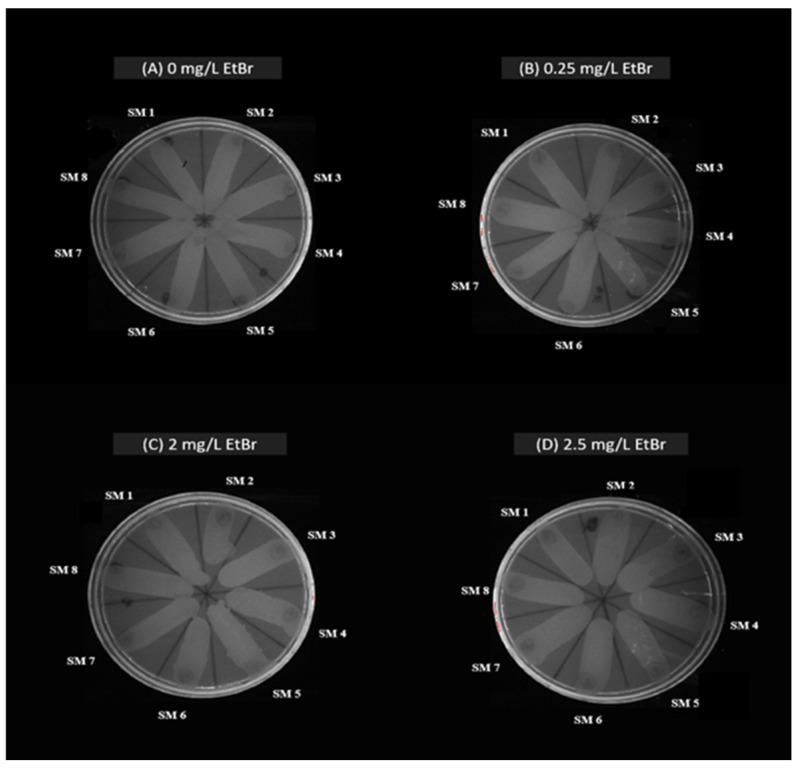
Efflux pump activity of *S. maltophilia* strains (SM 1 TO SM 15) obtained from canine samples. (**A**) 0 mg/L concentration of ethidium bromide; (**B**) 0.25 mg/L concentration of ethidium bromide; (**C**) 2 mg/L concentration of ethidium bromide; (**D**) 2.5 mg/L concentration of ethidium bromide.

**Figure 4 antibiotics-14-00639-f004:**
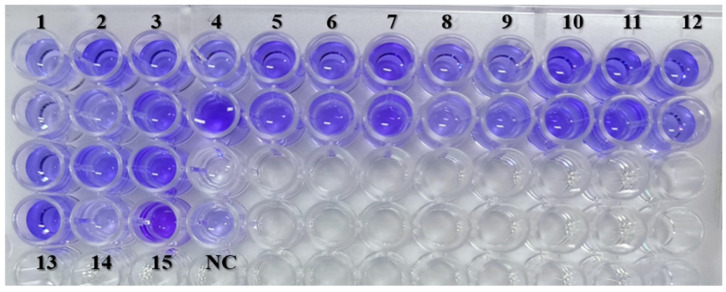
Microplate method to evaluate biofilm-forming ability. NC—Negative control; 1–15—Isolates SM1 to SM15.

**Figure 5 antibiotics-14-00639-f005:**
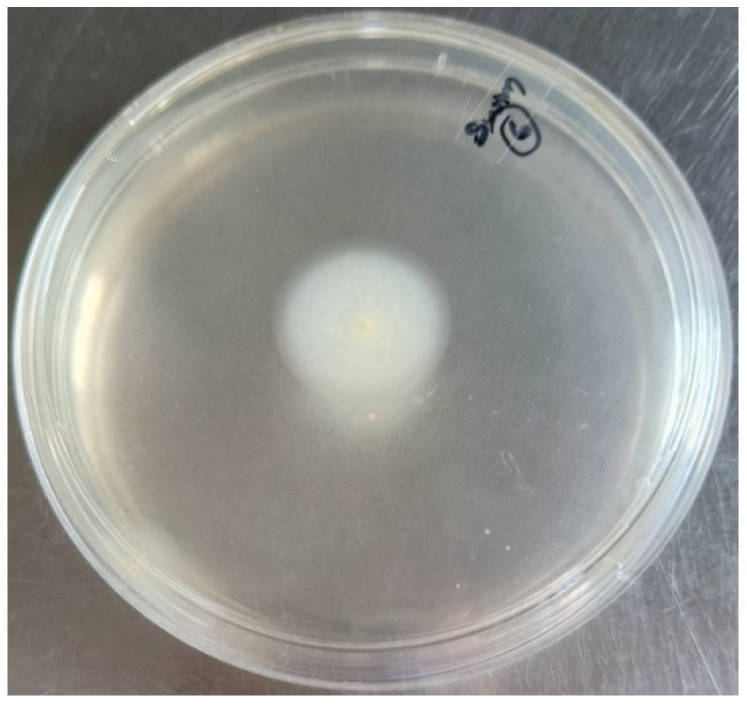
Swimming motility of *S. maltophilia* in modified LB media.

**Figure 6 antibiotics-14-00639-f006:**
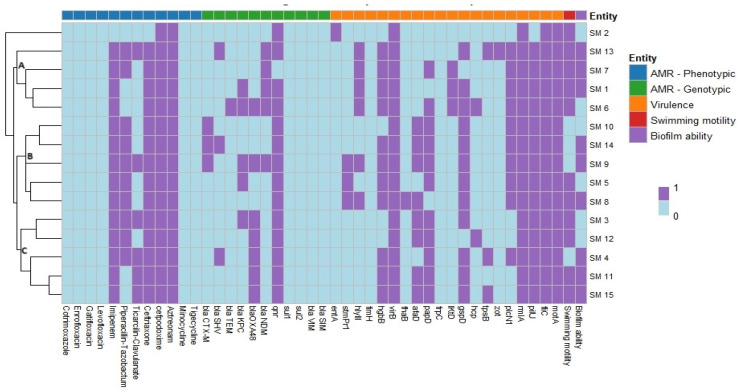
Heat map analysis with hierarchical clustering (dendrogram) of *S. maltophilia* isolates obtained from canine samples.

**Figure 7 antibiotics-14-00639-f007:**
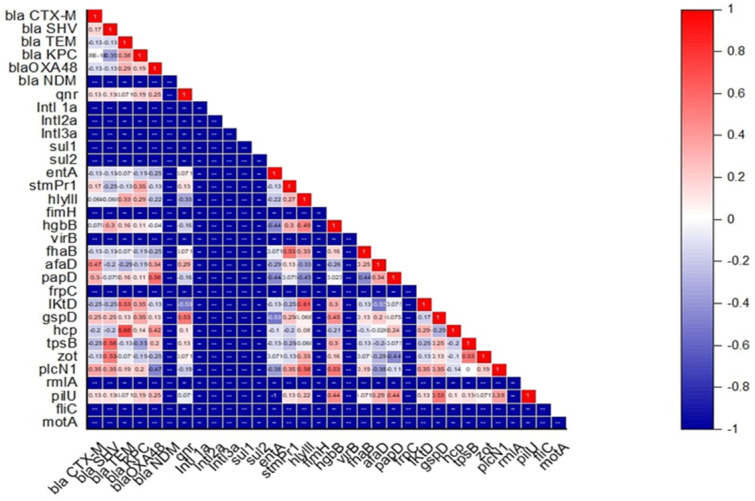
Correlation matrix analysis of antimicrobial resistance and virulence genes of *S. maltophilia* isolates obtained from canine samples.

**Table 1 antibiotics-14-00639-t001:** Identification of *S. maltophilia*.

Isolates	Date of Collection	Type of Sample	Gram’s Staining	Catalase Test	Oxidase Test	Glucose Fermentation Test	23S rRNA	MALDI-TOF SCORE
SM 1	12.01.24	Skin swab	Negative	Positive	Negative	Negative	Positive	2.25
SM 2	18.01.24	Skin swab	Negative	Positive	Negative	Negative	Positive	2
SM 3	21.01.24	Skin swab	Negative	Positive	Negative	Negative	Positive	2.06
SM 4	02.02.24	Skin swab	Negative	Positive	Negative	Negative	Positive	2.17
SM 5	07.02.24	Skin swab	Negative	Positive	Negative	Negative	Positive	2
SM 6	09.02.24	Skin swab	Negative	Positive	Negative	Negative	Positive	2.2
SM 7	17.02.24	Skin swab	Negative	Positive	Negative	Negative	Positive	2.26
SM 8	17.02.24	Skin swab	Negative	Positive	Negative	Negative	Positive	2.15
SM 9	19.02.24	Skin swab	Negative	Positive	Negative	Negative	Positive	2.23
SM 10	26.02.24	Skin swab	Negative	Positive	Negative	Negative	Positive	2.27
SM 11	11.03.24	Skin swab	Negative	Positive	Negative	Negative	Positive	2.26
SM 12	11.03.24	Skin swab	Negative	Positive	Negative	Negative	Positive	2.24
SM 13	13.03.24	Skin swab	Negative	Positive	Negative	Negative	Positive	2.26
SM 14	16.03.24	Skin swab	Negative	Positive	Negative	Negative	Positive	2.23
SM 15	16.03.24	Skin swab	Negative	Positive	Negative	Negative	Positive	2.17

**Table 2 antibiotics-14-00639-t002:** Characterization of the antimicrobial, virulence, and antibiofilm properties of *S. maltophilia*.

Strain Number	Phenotypic Resistance Pattern	AMR Genes	Virulence Genes	Biofilm Activity	Efflux Activity
SM 1	IPM, CTR, CPD, AT	*bla*_KPC_, *bla*_NDM_, *qnr*	*virB*, *motA*, *rmlA*, *fliC*, *pilU*, *gspD*, *hgbB*, *plcN1*, *hlyIII*, *lktD*	STRONG	ACTIVE
SM 2	CPD, AT	*qnr*	*virB*, *motA*, *rmlA*, *fliC*, *entA*	MODERATE	ACTIVE
SM 3	IPM, PIT, TCC, CTR, CPD, AT	*bla*_KPC_, *bla*_OXA-48_, *qnr*	*virB*, *motA*, *rmlA*, *fliC*, *pilU*, *gspD*, *papD*, *afaD*	MODERATE	ACTIVE
SM 4	IPM, PIT, TCC, CTR, CPD, AT	*bla*_SHV_, *bla*_OXA-48,_*qnr*	*virB*, *motA*, *rmlA*, *fliC*, *pilU*, *gspD*, *papD*, *hgbB plcN1*, *tpsB*	STRONG	ACTIVE
SM 5	IPM, PIT, CTR, CPD, AT	*bla*_KPC_, *qnr*	*virB*, *motA*, *rmlA*, *fliC*, *pilU*, *gspD*, *papD*, *hgbB*, *plcN1*, *stmPr1*	MODERATE	ACTIVE
SM 6	IPM, CTR, CPD, AT	*bla*_TEM_, *bla*_KPC_ *bla*_NDM_, *qnr*	*virB*, *motA*, *rmlA*, *fliC*, *pilU*, *gspD*, *papD*, *hgbB*, *plcN1*, *hlyIII*, *lktD*, *hcp*	MODERATE	ACTIVE
SM 7	IPM, PIT, CTR, CPD, AT	*bla* _NDM_	*virB*, *motA*, *rmlA*, *fliC*, *pilU*, *papD*, *hgbB*, *plcN1*, *hlyIII*, *lktD*	STRONG	ACTIVE
SM 8	IPM, PIT, CTR, CPD, AT	*qnr*	*virB*, *motA*, *rmlA*, *fliC*, *pilU*, *gspD*, *hgbB*, *plcN1*, *afaD*, *hlyIII*, *fhaB*, *stmPr1*	STRONG	ACTIVE
SM 9	IPM, PIT, TCC, CTR, CPD, AT	*bla*_CTX-M_, *bla*_KPC_, *bla*_OXA-48_, *bla*_NDM_, *qnr*	*virB*, *motA*, *rmlA*, *fliC*, *pilU gspD*, *papD*, *hgbB plcN1*, *afaD*, *hlyIII*, *stmPr1*	STRONG	ACTIVE
SM 10	IPM, PIT, CTR, CPD, AT	*bla*_CTX-M_, *qnr*	*virB*, *motA*, *rmlA*, *fliC*, *pilU*, *gspD*, *papD*, *plcN1*, *afaD*	MODERATE	ACTIVE
SM 11	IPM, TCC, CTR, CPD, AT	*bla*_OXA-48_, *qnr*	*virB*, *motA*, *rmlA*, *fliC*, *pilU*, *gspD*, *papD*, *hgbB*, *afaD*	STRONG	ACTIVE
SM 12	IPM, PIT, CTR, CPD, AT	*bla*_OXA-48_, *qnr*	*virB*, *motA*, *rmlA*, *fliC*, *pilU*, *papD*, *afaD*, *hcp*	MODERATE	ACTIVE
SM 13	IPM, PIT, TCC, CTR, CPD, AT	*bla*_SHV_, *bla*_NDM_, *qnr*	*virB*, *motA*, *rmlA*, *fliC*, *pilU*, *gspD*, *hgbB*, *plcN1*, *hlyIII*, *tpsB*, *zot*	STRONG	ACTIVE
SM 14	IPM, PIT, CTR, CPD, AT	*bla*_CTX-M_, *bla*_SHV_, *qnr*	*virB*, *motA*, *rmlA*, *fliC*, *pilU*, *gspD*, *papD*, *hgbB*, *plcN1*, *afaD*	STRONG	ACTIVE
SM 15	IPM, TCC, CTR, CPD, AT	*qnr*	*virB*, *motA*, *rmlA*, *fliC*, *pilU*, *gspD*, *papD*, *hgbB*, *afaD tpsB*	STRONG	ACTIVE

IPM—imipenem, CTR—ceftriaxone, CPD—cefpodoxime, AT—aztreonam, PIT—piperacillin-tazobactam, TCC—ticarcillin-clavulanate.

## Data Availability

All data supporting the findings of this study are contained within this manuscript and will be made available upon request.

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
