# Peer review of "First Report of Stenotrophomonas maltophilia from Canine Dermatological Infections: Unravelling Its Antimicrobial Resistance, Biofilm Formation, and Virulence Traits"

_antibiotics, 2025, doi:10.3390/antibiotics14070639_

Round 1

Reviewer 1 Report

Comments and Suggestions for Authors

The manuscript presents an eye-opening report on Stenotrophomonas maltophilia isolated from canine dermatological infections, providing novel insights into its antimicrobial resistance, biofilm-forming ability, and virulence characteristics.

However, several issues merit revision and clarification. In the Results section under “Characterization of antimicrobial resistance in S. maltophilia,” I have concerns regarding Figure 2. The PCR band positions appear inconsistent—for example, lane 2 (817 bp) appears lower than lane 3 (798 bp), which raises questions about gel accuracy or labeling. Additionally, the figure legend contains repetitive phrasing ("S. maltophilia isolate positive for...") which may confuse readers; it would be clearer to list the specific AMR genes associated with visible bands.

Similarly, for Figure 5, the legend should be revised to eliminate redundancy. Clarifying the exact band size (e.g., 431 bp) with visual indicators such as an arrowhead would improve interpretability.

In the Discussion, the text addressing antimicrobial resistance genes (ARGs) and biofilm production lacks comparative context. I recommend integrating findings from previous studies to better position your results within the existing literature. Moreover, although study limitations are noted, additional limitations could be acknowledged to provide a more balanced perspective.

In the Methods section, the description of the biofilm formation assay requires further detail. It is important to specify how biofilm strength categories (e.g., strong, moderate) were determined. Furthermore, the methodology for confirming the identity of PCR products should be described, especially given the absence of a reference S. maltophilia strain harboring each target gene in this study.

Addressing these concerns will strengthen the manuscript’s scientific clarity and rigor.

Author Response

In the Results section under “Characterization of antimicrobial resistance in S. maltophilia,” I have concerns regarding Figure 2. The PCR band positions appear inconsistent—for example, lane 2 (817 bp) appears lower than lane 3 (798 bp), which raises questions about gel accuracy or labeling.

The labelling error in lane 2, 3, and 7 has been rectified. Please see Figure 2.

Additionally, the figure legend contains repetitive phrasing ("S. maltophilia isolate positive for...") which may confuse readers; it would be clearer to list the specific AMR genes associated with visible bands.

Agreed and complied. Please see Figure 2.

Similarly, for Figure 5, the legend should be revised to eliminate redundancy. Clarifying the exact band size (e.g., 431 bp) with visual indicators such as an arrowhead would improve interpretability.

Agreed and complied. Please see Figure 5.

In the Discussion, the text addressing antimicrobial resistance genes (ARGs) and biofilm production lacks comparative context. I recommend integrating findings from previous studies to better position your results within the existing literature.

We appreciate the reviewer's valuable suggestion regarding the integration of comparative findings. While this represents the first investigation of S. maltophilia specifically from canine dermatological sources, we acknowledge that our findings can be meaningfully contextualized within the broader literature on S. maltophilia from various sources.

We have incorporated these comparative perspectives into our Discussion section to better position our findings within the existing body of knowledge while emphasizing the novel contribution of documenting these characteristics in canine dermatological isolates. Please see lines 215-224, 237-239, 241-243 and 298-303.

Moreover, although study limitations are noted, additional limitations could be acknowledged to provide a more balanced perspective.

We thank the reviewer for highlighting the need to elaborate on study limitations. As the first global report of S. maltophilia in canine dermatological infections, our study presents baseline prevalence and phenotypic-genotypic characterization. However, we acknowledge several limitations inherent to pioneering research. The 5% prevalence (15/300) was derived from a single veterinary teaching hospital over a three-month period, which may not reflect broader geographic or seasonal trends. The absence of standardized protocols for isolating S. maltophilia from canine skin, and limited comparative data, posed methodological challenges requiring adaptation of existing techniques.

Furthermore, while we optimized PCR using a reference S. maltophilia strain (MCC 2083), we lacked positive controls for individual resistance and virulence genes and did not perform sequence confirmation, which limits molecular specificity. Our focus was on detection and characterization, not clinical correlation; thus, associations with treatment outcomes or disease severity remain unexplored.

Despite these limitations, this study provides foundational data for S. maltophilia in canine dermatology, highlights its antimicrobial resistance and virulence potential, and supports the need for expanded surveillance under the One Health framework.

In the Methods section, the description of the biofilm formation assay requires further detail. It is important to specify how biofilm strength categories (e.g., strong, moderate) were determined.

Biofilm strength categorization has been added in the respective section. Please see line 365-367.

Furthermore, the methodology for confirming the identity of PCR products should be described, especially given the absence of a reference S. maltophilia strain harboring each target gene in this study.

In this study, we used a reference strain of S. maltophilia (MCC 2083) to optimize the PCR targeting 16S rRNA gene specific for S. maltophilia. However, due to the unavailability of reference strains harboring each specific virulence and antimicrobial resistance gene, we did not use positive controls for those gene targets. To ensure specificity, we used primer sequences published earlier and confirmed the expected amplicon sizes.

Reviewer 2 Report

Comments and Suggestions for Authors

Dear Authors,

Congratulation for your work ! A lot of data are presented in this paper.

I have a few questions:

  1. Do you have more information about the dogs (age, sex, breed, intercurent diseases)?
  2. Can you tell me what is the prevalence of this pathogen in animals and humans?
  3. Why did you choose this pathogen?

And a suggestion:

You refer to Table 1 (see line 86), but on the next page, is there anything else? I suggest moving Table 1 to the next page (page 3).

Could you give more details about the interpretation of the MALDI-TOF results? 

Author Response

Do you have more information about the dogs (age, sex, breed, intercurent diseases)?

Yes, the information such as age, sex, breed, clinical condition and treatment given to the dogs from which the samples were taken has been submitted as supplementary table to the Assistant editor along with ARRIVE guidelines checklist.

Can you tell me what is the prevalence of this pathogen in animals and humans?

The prevalence of S. maltophilia in humans globally, has increased from 7% in 2004–2007 to 15% in 2020–2022 which accounts for 32.2% from surgical intensive care unit (ICU) patients followed by the patients with cardiac problems (29.8%) and the paediatric ICU (10%) (AlFonaisan et al., 2024). [https://doi.org/10.1038/s41598-024-65509-z]

On the other hand, only scarce data was available on the prevalence status of                                    S. maltophilia in animals. As mentioned in the manuscript, some reports of S. maltophilia in different species of animals with chronic respiratory disease and urinary infections, including horses, cats, dogs, and ball pythons have been noted. (Albini et al., 2009; Kralova-Kovarikova et al., 2012). Further, only one study in India has reported its occurrence in animals, specifically in sarcoptic pig skin ulcers (Thenissery et al., 2022), with no prior studies in dogs.

Why did you choose this pathogen?

Stenotrophomonas maltophilia is an emerging opportunistic pathogen in both human and veterinary medicine. While it is well recognized in hospital-acquired infections in humans, its role in animal infections, particularly dermatological conditions, is underexplored. Studying this bacterium in canine skin infections is critical for assessing potential zoonotic transmission along with its virulence, antimicrobial resistance and biofilm forming ability and in forming One Health surveillance strategies.

You refer to Table 1 (see line 86), but on the next page, is there anything else? I suggest moving Table 1 to the next page (page 3).

Agreed and complied. See page 3.

Could you give more details about the interpretation of the MALDI-TOF results?

A pure bacterial colony is smeared onto a MALDI target slide, and a matrix solution is added. The matrix helps to ionize the bacterial proteins, which are then separated and detected by the MALDI-TOF MS instrument. The instrument generates a mass spectrum, which is a unique profile of the bacterial proteins, especially ribosomal proteins. This PMF serves as the fingerprint of the bacteria. The generated PMF is compared to a database of known bacterial species. The software calculates a match score based on the similarity between the unknown PMF and the reference spectra. A higher match score indicates a stronger match to a specific bacterial species. Generally, scores above 2.0 are considered highly reliable for species-level identification. Scores below 1.7 are considered unreliable. Scores between 1.7 and 2.0 may indicate a probable genus-level identification, while scores between 2.0 and 2.3 may indicate probable species-level identification.

Reviewer 3 Report

Comments and Suggestions for Authors

In this study, the authors report that 15 Stenotrophomonas maltophilia isolates were recovered from skin swabs of dogs presenting with dermatological conditions. While the study provides preliminary data on the occurrence of multidrug-resistant S. maltophilia in canine skin infections (reported at the Teaching Veterinary Hospital, Madras Veterinary College, India), the results do not adequately establish causality or clinical significance. The primary evidence presented as growth of the bacteria on media supplemented with antibiotics does not sufficiently demonstrate virulence or pathogenic potential. The authors are encouraged to include sequencing data to strengthen the strain identification and clonal linkage claims. The authors are also encouraged to provide additional clarifications as outlined below.

  • MALDI-TOF Identification

The authors have used MALDI-TOF MS for rapid bacterial identification. They are encouraged to submit the raw mass spectra or spectral maps for all bacterial strains as supplementary data. Additionally, the Results section should clearly state whether a reference strain was used to identify all isolates. If so, the authors should specify which strain was used and provide a justification for this approach. If no reference strain was used, a rationale should also be included.

  • Figure Legends and other Formatting

As a formatting note, figure legends are typically left-aligned, not centered. Centered legends may reduce readability in both manuscripts and final publications. Additionally, inconsistencies in font type and formatting are observed throughout the manuscript. The authors should ensure uniform font usage and adhere to the journal’s formatting guidelines.

  • Figure 2: Gene Amplification Gel

Figure 2 is currently unclear. The authors show PCR amplification of seven genes, with reference to various n per well. It is unclear what these n values represent. Are they numbers of skin swabs, biological replicates, or pooled DNA samples? The authors should clearly state whether the gel image shows representative lanes from one positive sample per gene, or a pooled analysis. This clarification is essential to interpret the gel results accurately.

  • Figure 3: Efflux Pump Activity

The results in Figure 3 show no apparent difference in efflux pump activity across EtBr concentrations from 0 to 2.5 mg/L. If available, authors could consider including quantitative data or statistical comparisons to support this conclusion. The same applies for Figure 6 representing swimming motility of S. maltophilia in modified LB media. If available, authors could consider including quantitative data or statistical comparisons to support this conclusion.

  • Figure 4: Virulence Gene Profile

Figure 4 lacks clarity in interpretation. The blue dye in the wells does not specify what the colors represent (e.g., intensity, presence/absence, expression levels). It is recommended that the authors quantify the virulence gene data, perhaps as a bar graph or annotated heatmap. In addition, the figure legend should describe the meaning of each color shade and explain how the data were derived.

  • Figure 5: Gel Image Concerns

Figure 5 raises concerns regarding marker interpretation and data consistency. For instance, Lane 3 shows a band labeled 1235 bp migrating near the 1779 bp marker, while Lane 11 shows a 1239 bp band that appears to run much faster, which is inconsistent. These discrepancies raise questions about gel accuracy, ladder annotation, or labeling errors. Furthermore, the authors should clarify the relevance of this gel to the central story of the paper. If it does not directly contribute to the findings, reconsidering its inclusion or improving the explanation is advised.  

Author Response

While the study provides preliminary data on the occurrence of multidrug-resistant S. maltophilia in canine skin infections (reported at the Teaching Veterinary Hospital, Madras Veterinary College, India), the results do not adequately establish causality or clinical significance.

The current study aimed to document the occurrence of S. maltophilia in dogs with dermatological ailments which was not reported earlier. Further the study explored antimicrobial resistance, virulence and biofilm forming ability of the obtained isolates and corresponding results have been recorded. While these findings provide valuable baseline data, further research is required to establish causality and determine the clinical significance of S. maltophilia in canine dermatological infections.

The primary evidence presented as growth of the bacteria on media supplemented with antibiotics does not sufficiently demonstrate virulence or pathogenic potential.

We agree that growth on antibiotic-supplemented media is primarily recommended for isolation of S. maltophilia. In this view, our study utilized media containing Vancomycin, Imipenem, Amphotericin B supplement which will selectively favour growth of S. maltophilia and inhibits the growth of another competing microflora.   Further, the pathogenic potential of the isolates was demonstrated through detection of virulence genes, motility assay and biofilm assay.

The authors are encouraged to include sequencing data to strengthen the strain identification and clonal linkage claims.

While we agree that sequencing data would provide additional resolution, the primary objective of our study was not to characterize the strains at the genomic level or assess clonal relationships, but rather to determine the prevalence, virulence and antimicrobial resistance attributes of S. maltophilia. Also, sequencing-based strain typing was beyond the intended scope and resources of this study. Further, identification was done by employing MALDI-TOF MS

The authors have used MALDI-TOF MS for rapid bacterial identification. They are encouraged to submit the raw mass spectra or spectral maps for all bacterial strains as supplementary data.

We appreciate your valuable suggestion. In our study, MALDI-TOF MS was employed as a rapid and reliable method for species-level identification of Stenotrophomonas maltophilia isolates, following the standardized ethanol–formic acid extraction protocol. All isolates yielded identification scores greater than 2.0, which, according to Bruker Biotyper criteria, indicate high-confidence species-level identification. The method is widely validated and routinely used in both clinical and veterinary microbiology for bacterial diagnostics.

Due to platform constraints and proprietary limitations associated with the MALDI Biotyper software and database, we are unable to provide the raw mass spectra or spectral maps as supplementary data. However, the complete MALDI-TOF score data are provided in Table 1 of the manuscript, and all identifications were internally validated against a reference spectrum.

Additionally, the Results section should clearly state whether a reference strain was used to identify all isolates. If so, the authors should specify which strain was used and provide a justification for this approach. If no reference strain was used, a rationale should also be included.

In this study, we used a reference strain of S. maltophilia (MCC 2083) to optimize the PCR targeting 16S rRNA gene specific for S. maltophilia. However, due to the unavailability of reference strains harboring each specific virulence and antimicrobial resistance gene, we did not use positive controls for those gene targets. To ensure specificity, we used primer sequences published earlier and confirmed the expected amplicon sizes. Please see line 320-321.

As a formatting note, figure legends are typically left-aligned, not centered. Centered legends may reduce readability in both manuscripts and final publications.

Agreed and complied. Please see Figures 1, 2, 3, 4 and 5.

Additionally, inconsistencies in font type and formatting are observed throughout the manuscript. The authors should ensure uniform font usage and adhere to the journal’s formatting guidelines.

Agreed and complied.

Figure 2 is currently unclear. The authors show PCR amplification of seven genes, with reference to various n per well. It is unclear what these n values represent. Are they numbers of skin swabs, biological replicates, or pooled DNA samples? The authors should clearly state whether the gel image shows representative lanes from one positive sample per gene, or a pooled analysis. This clarification is essential to interpret the gel results accurately.

The gel image was assembled to present the amplification of seven different genes, with each lane corresponding to an individual sample (not a pooled analysis) that tested positive for the respective gene. Some samples harboured more than one gene, which led to the appearance of multiple bands in each lane.

The results in Figure 3 show no apparent difference in efflux pump activity across EtBr concentrations from 0 to 2.5 mg/L. If available, authors could consider including quantitative data or statistical comparisons to support this conclusion. The same applies for Figure 6 representing swimming motility of S. maltophilia in modified LB media. If available, authors could consider including quantitative data or statistical comparisons to support this conclusion.

In the current study, assessment of efflux pump activity using ethidium bromide was performed as a qualitative assay to determine antibiotic resistance. In a similar fashion, the motility pattern was confirmed in modified LB media which showcased swimming motility among the isolates tested. 

Figure 4 lacks clarity in interpretation. The blue dye in the wells does not specify what the colors represent (e.g., intensity, presence/absence, expression levels).

The intensity of the dye is directly proportional to the amount of biofilm formation by each isolate which is measured through microplate reader at 492 nm and expressed as Optical density (OD). Biofilm strength was categorized based on OD values as Negative: < Ac; Weak: Ac < A ≤ 2Ac; Moderate: 2Ac < A ≤ 4Ac and Strong: A > 4Ac, where Ac – OD Value of Negative control; A – OD Value of Sample. Please see line 365.

It is recommended that the authors quantify the virulence gene data, perhaps as a bar graph or annotated heatmap. In addition, the figure legend should describe the meaning of each color shade and explain how the data were derived.

We thank the reviewer for this valuable suggestion. In response, we have incorporated a heatmap (Figure 7) with hierarchical clustering to visualize the distribution and frequency of virulence genes across the S. maltophilia isolates. This graphical representation complements the tabulated PCR results and provides a clearer overview of isolate-specific virulence profiles. The figure legend shows two colour shades indicating presence or absence of the various entities among the tested isolates, such as AMR-phenotypic, AMR- Genotypic, virulence, swimming motility and biofilm ability.

Figure 5 raises concerns regarding marker interpretation and data consistency. For instance, Lane 3 shows a band labeled 1235 bp migrating near the 1779 bp marker, while Lane 11 shows a 1239 bp band that appears to run much faster, which is inconsistent. These discrepancies raise questions about gel accuracy, ladder annotation, or labeling errors

Upon review, we identified that the discrepancies were due to labeling errors during figure preparation. We have corrected the annotations to accurately reflect the band sizes and marker positions. The revised figure now ensures consistency between the reported amplicon sizes and their actual migration patterns. Please see figure 5.

Furthermore, the authors should clarify the relevance of this gel to the central story of the paper. If it does not directly contribute to the findings, reconsidering its inclusion or improving the explanation is advised.

Thank you for the insightful comment. The gel image in question (Figure 5) depicts the PCR amplification of key virulence genes detected in S. maltophilia isolates. This figure is central to the study's objective of characterizing the virulence potential of the isolates, which complements the antimicrobial resistance and biofilm formation data.

To enhance clarity, we have revised the figure legend to explicitly state which virulence genes are shown and their respective amplicon sizes. We also elaborated in the Results section on how these gene targets relate to the phenotypic traits (e.g., motility, biofilm production) and their overall contribution to the pathogen’s virulence profile.

We believe this figure provides critical visual validation of molecular data supporting the study’s One Health perspective on the zoonotic potential of S. maltophilia and have therefore retained it in the revised manuscript.